# Physical fighting among Egyptian adolescents: social and demographic correlates among a nationally representative sample

Karen L. Celedonia[1], Michael L. Wilson[1], Hanan A. El Gammal[2] and Abeer M. Hagras[3]

[1] Centre for Injury Prevention and Community Safety, PeerCorps Trust Fund, Dar es Salaam, Tanzania
[2] Family Medicine Department, Faculty of Medicine, Suez Canal University, Ismailia, Egypt
[3] Department of Forensic Medicine and Clinical Toxicology, Faculty of Medicine, Suez Canal University, Ismailia, Egypt

Corresponding author
Karen L. Celedonia,
karen.celedonia@peercorpstrust.org

## ABSTRACT

**Introduction.** Adolescent interpersonal violence is a global public health problem, yet gaps remain in the epidemiologic literature on adolescent violence in low- and middle-income countries (LMIC). Prevalence rates and risk and protective factors reported in high-income countries may be different from those reported in LMICs. Culturally-relevant epidemiologic data is important in efforts aimed at addressing adolescent interpersonal violence in these countries.

**Methods.** A cross-sectional study of Egyptian adolescent involvement in violent behavior was conducted. Data collected from a 2006 school-based survey initiative were used; participants were adolescents aged 11–17 ($N = 5,249$). Some participants were excluded from the dataset due to incomplete data ($N = 111$) resulting in a final sample of 5,138. Bivariate and logistic regression analyses were run to determine demographic and social variables associated with participation in physical fighting.

**Results.** Thirty-one percent of adolescents reported being involved in a physical fight. Previously reported risk factors for violent behavior among adolescents such as depressive symptoms ($OR = 1.29$; $CI = 1.11–1.50$) and bullying victimization ($OR = 2.44$; $CI = 2.12–2.83$) were positively associated with violent behavior in the present study, while the more novel factor of sedentary behavior was also observed as having a positive association with violent behavior ($OR = 1.43$; $CI = 1.21–1.69$). Known protective factors such as helpful peers ($OR = 0.75$; $CI = 0.62–0.90$) and understanding parents ($OR = 0.67$; $CI = 0.56–0.81$) were found to have negative associations with violent behavior in the present study, in addition to the counterintuitive protective effect of having fewer friends ($OR = 0.75$; $CI = 0.60–0.92$).

**Conclusions.** Prevalence rates of adolescent interpersonal violence in Egypt are similar to rates in other LMICs. The high reported rates of depressive symptomatology and bully victimization along with their positive association with physical fighting suggest that interventions aimed at treating and preventing these problems may help mitigate the likelihood of adolescents engaging in violent behavior; involvement in appropriate physical activity in a safe environment may be beneficial as well. More research is needed to understand the observed protective factor of having fewer friends.

## INTRODUCTION

Interpersonal violence among adolescents has become an increasingly important global public health concern in recent years (*Sugimoto-Matsuda & Braun, 2013*). Nearly one third of adolescents from North America and Europe reported being in a physical fight within a year to a year and a half prior to being surveyed (*Eaton et al., 2012*; *Walsh et al., 2013*), and in the Middle East and Sub-Saharan Africa, more than half of adolescents report involvement in physical fighting (*Gofin, Palti & Mandel, 2000*; *Rudatsikira et al., 2007*). Prevalence rates of physical fighting and other forms of interpersonal violence in low- and middle-income countries–particularly those experiencing social and political unrest–remain elevated and, in some instances, are increasing (*Krug et al., 2002*). In response to this concerning trend, there has been a call for concerted public health efforts to reduce violence among adolescents worldwide (*Krug et al., 2002*; *OECD, 2011*).

One component of these efforts has included research aimed at discerning individual and social factors associated with adolescent involvement in violent behavior. From this research, several common risk factors have been determined, among them including drug, alcohol or tobacco use; poor behavior control; emotional distress; and poor parental supervision/involvement (*DHHS, 2001*). The majority of this research, however, has been conducted with adolescent samples from high-income countries leaving important knowledge gaps in low- and middle-income country settings (LMIC).

Additionally, not all adolescents exhibit violent behavior, indicating that there may be other, micro-level factors that are exerting influence either as risk or protective factors. A better understanding of individual and social correlates of violence among adolescents in LMICs is important to violence prevention in these countries. With rates of non-fatal injuries resulting from violent behavior starting to increase during mid-adolescence (age 14 years) and violence peaking during late adolescence (*Odero & Kibosia, 1995*; *Butchart, Kruger & Nell, 1997*; *DHHS, 2001*), violence prevention programs geared towards adolescents are critical to protecting the well-being of this demographic.

The present study reports the results of analyses conducted on data collected from a school-based survey administered to Egyptian adolescents. Epidemiologic data on individual and social correlates of physical violence in Egyptian adolescents is limited (*Youssef, Attia & Kamel, 1999*; *Ez-Elarab et al., 2007*). We sought to add to the existing literature on youth violence in Egypt by examining physical fighting among Egyptian adolescents. The aims of the study were twofold: (1) to determine which individual and social factors were associated with an increased likelihood of adolescents participating in physical fights and (2) whether the presence of known protective factors mediated involvement in physical fighting.

## METHODS

### Sample

The data for this study were derived from the 2006 Egypt contribution to the Global School-based Health Survey (GSHS). The GSHS was developed by the World Health Organization in collaboration with the U.S. Centers for Disease Control. It collects relevant information for the discernment of behavioral and health risks among adolescents of school age in more than 40 mainly LMICs. Detailed information on the data collection methods, questionnaire and procedures may be found elsewhere (http://www.cdc.gov/gshs/). Briefly, a two-stage cluster sampling design was used to facilitate the collection of data representing all students in Preparatory first, second, and third grade in Egypt. At stage one, schools were selected with a probability proportional to their enrollment size. At stage two, classrooms within the selected schools were randomly selected and all students in selected classes were eligible to participate. A total of 5,249 students aged 11–17 years participated in the Egyptian GSHS.

The school response rate was 100% and the student response rate was 87%. The response rate of 87% was high due to being carried out in a controlled environment (schools) among a captive audience (school-attending adolescents). In a global comparison of 34 countries using the same methodology and questionnaire, the average response rate we calculated at 84.14 (SD 8.39) based on the rates presented in the study. All students were informed of the anonymous nature of the questionnaire and were free to participate. Answers to each of the completed questionnaires were self-reported on a computer scannable answer sheet. Other than verifying heights and weights, there were no validation measures used for the other responses in the survey, including the responses to items used for the present study. At the time of data collection, the research committee of the Ministry of Education approved the study including the questionnaire.

In the present study, we excluded 111 participants that did not have complete information on age and sex which resulted in a final sample of 5,138 (48% female).

### Measurements

We derived the dependent variable, physical fighting, from one question in the GSHS: "*During the past 12 months, how many times were you in a physical fight?*" Response options ranged from "*0 times*", "*1 time*", "*2 or 3 times*", "*4 or 5 times*", "*6 or 7 times*", "*8 or 9 times*", "*10 or 11 times*" or "*12 or more times*". A physical fight was defined as "when two or more students of about the same strength and power choose to fight each other". For the purpose of our analyses, participants were classified as having participated in a physical fight if they reported being in two or more fights ($N = 1,593$). If one or fewer fights were reported, participants were classified as not participating in a physical fight ($N = 3,545$).

We investigated five independent variables at the individual level (anxiety, signs of depression, truancy, physical activity, and bullying victimization) and four independent variables at the social level (presence of supportive parental figures, presence of helpful peers, extent of social network, and food insecurity). Details on how these variables were created can be found in Table 1.

**Table 1  Independent variable derivation from GSHS survey data (2006).**

| Survey question | Coding | Variable |
|---|---|---|
| **Individual-level variables** | | |
| How old are you? | 11–17 years (coded categorically) | Age |
| What is your sex? | Male (1)<br>Female (0) | Sex |
| During the past 12 months, how often have you been so worried about something that you could not sleep at night? | Most of the time/always (1)<br>Never/rarely/sometimes (0) | Anxiety |
| During the past 12 months, did you ever feel so sad or hopeless almost every day for *two weeks or more in a row* that you stopped doing your usual activities? | Yes (1)<br>No (0) | Signs of depression |
| During the past 12 months, how often have you felt lonely? | Most of the time/always (1)<br>Never/rarely/sometimes (0) | Loneliness |
| During the past 30 days, how many days did you miss classes or school without permission? | 0–2 times (0)<br>3 to or more days (1) | Truancy |
| During the past 30 days, on how many days were you bullied? | 0 times (0)<br>1 or more times (1) | Bullying victimization |
| During the past 7 days, on how many days were you physically active for a total of at least 60 min per day? | 3 days or less (0)<br>4 days or more (1) | Physical activity |
| How much time do you spend during a *typical or usual* day sitting and watching television, playing computer games, talking with friends, or doing other sitting activities? | 2 h or less (0)<br>3 h or more (1) | Sedentary |
| During the past 12 months, how many times were you in a physical fight? | 0–1 times (0)<br>2 or more times (1) | In a fight |
| **Social-level variables** | | |
| During the past 30 days, how often did your parents or guardians understand your problems and worries? | Most of the time/always (1)<br>Never/rarely/sometimes (0) | Supportive parental figures |
| During the past 30 days, how often were most of the students in your school kind and helpful? | Always (1)<br>Most of the time/<br>Never/rarely/sometimes (0) | Helpful peers |
| How many close friends do you have? | 0 close friends (0)<br>1 close friends (1)<br>2 close friends (2)<br>3+ close friends (3) | Close friends |
| During the past 30 days, how often did you go hungry because there was not enough food in your home? | Most of the time/always (1)<br>Never/rarely/sometimes (0) | Food insecurity |

## Statistical analysis

We first examined the distribution of selected independent variables within the dichotomized physical fighting variable. Differences between physical fighting involvement among the variables were screened for statistical significance using Pearson's chi-square for categorical variables and the t-test for continuous variables (age). We then created two binary logistic regression models. These were intended to model the ability of the selected independent variables to predict the dichotomized physical fighting variable. The first model included all variables which were significant at the bivariate level. A second model

**Table 2 Distribution of selected factors according to categories of physical fighting among school-attending adolescents in Egypt (2006).**

| Variable | Not involved in physical fights ($n = 3,545$) | Involved in physical fights ($n = 1,593$) | $p$-value | Chi-square or $t$-value |
|---|---|---|---|---|
| Age (mean) | 13.2 (0.93) | 13.2 (0.99) | 0.410 | −0.825 |
| Sex (female) | 55.0 | 33.1 | <0.001 | 211.606 |
| Food insecurity | 5.0 | 6.2 | 0.066 | 3.400 |
| Bullied | 45.6 | 70.3 | <0.001 | 269.243 |
| Loneliness | 8.4 | 13.1 | <0.001 | 27.529 |
| Anxiety | 8.0 | 14.1 | <0.001 | 44.659 |
| Signs of depression | 30.6 | 34.2 | <0.011 | 6.492 |
| Truancy | 14.0 | 21.9 | <0.001 | 49.386 |
| Social network | | | | |
|     No close friends | 7.3 | 6.2 | <0.001 | 21.134 |
|     One close friend | 17.8 | 13.1 | — | — |
|     Two close friends | 30.3 | 31.2 | — | — |
|     Three or more close friends | 44.6 | 50.0 | — | — |
| Helpful peers | 27.1 | 16.8 | <0.001 | 64.074 |
| Sedentary | 21.1 | 31.5 | <0.001 | 63.605 |
| Physically active | 18.0 | 20.4 | 0.044 | 4.072 |
| Understanding parents | 19.0 | 11.7 | <0.001 | 41.973 |

**Notes.**
All variables are expressed as proportions (in %) except for age (mean and standard deviation).

adjusted only for age and sex. We reported the measures of association as odds ratios (OR) accompanied by 95% confidence intervals (CI). All analyses were carried out using Stata 12 (*StataCorp, 2011*).

## RESULTS

Within the recall period, 31% ($n = 1,593$) of participants reported being involved in two or more physical fights, a majority of which were males (67%). The mean age of the sample was 13.2 years old (SD = 0.95). Thirty-two percent of respondents reported signs of depression, with 10% indicating loneliness and 10% percent anxiety. Food insecurity was found in 5.4% of the sample and more than half (53%) of all participants reported being bullied during three or more days during the recent 30-day period.

Table 2 shows the crude distribution of selected factors according to physical fighting category. The bivariate analyses revealed that within all but three of the selected variables, significant differences existed between participants who had been involved in physical fights and those who were not. No significant differences existed with regard to food insecurity among the physical fighting categories or physical activity.

Table 3 shows the distribution of variables by sex. The age and sex adjusted analysis (Table 4) revealed significant associations for all selected variables with the exception of age, social network variables and physical activity. As in the model which adjusted for all covariates, female sex (OR = 0.41; CI = 0.36–0.46) and having helpful peers (OR = 0.63; CI = 0.54–0.74) were associated with fewer reports of fighting activity.

**Table 3 Distribution of selected variable categories by sex among a nationally representative sample of Egyptian adolescents (2006).**

| Variable | Male | Female | Chi-square | p-value |
|---|---|---|---|---|
| Physical fight | 40.1 | 21.3 | 211.606 | <0.001 |
| Food insecurity | 5.3 | 5.4 | 0.003 | 0.958 |
| Bullied | 54.0 | 52.5 | 1.190 | 0.275 |
| Lonely | 7.7 | 12.0 | 26.677 | <0.001 |
| Anxiety | 9.7 | 10.2 | 0.382 | 0.536 |
| Signs of depression | 29.3 | 34.4 | 15.062 | <0.001 |
| Number of close friends | | | | |
|     None | 5.4 | 8.6 | 104.628 | <0.001 |
|     One | 12.4 | 20.5 | — | — |
|     Two | 30.4 | 30.8 | — | — |
|     Three or more | 51.9 | 40.0 | — | — |
| Truancy | 18.6 | 14.1 | 18.375 | <0.001 |
| Helpful peers | 22.7 | 25.2 | 4.381 | 0.036 |
| Understanding parents | 15.3 | 18.4 | 8.682 | 0.003 |
| Physical activity | 26.5 | 10.5 | 216.478 | <0.001 |

**Notes.**
Percentages of the total within each category are listed.

Having understanding parents (OR = 0.67; CI = 0.56–0.81) was found to be significantly protective. None of the social network variables were found to be associated with fighting behavior. As in the previous model being bullied, loneliness, anxiety, signs of depression, truancy and sedentary behavior were associated with higher rates of fighting.

After adjusting for all associated covariates in the first model (Table 5), and compared to those who did not report being involved in physical fighting, those who had been involved in physical fights were younger (OR = 0.90; CI = 0.84–0.97) and less likely to be male (OR = 0.38; 0.32–0.43). Being bullied was significantly associated with an increase in fighting involvement (OR = 2.44; CI = 2.12–2.83) as was missing days of school (OR = 1.26; CI = 1.05–1.52). Those who reported signs of depression were also more likely to have reported being involved in fighting behavior (OR = 1.29; CI = 1.11–1.50). Having fewer close friends appeared to indicate an effect which was protective, but the result was only statistically significant for those who reported having only one friend (OR = 0.75; CI = 0.60–0.92). Having helpful peers in school (OR = 0.75; CI = 0.62–0.90) had a protective effect while sedentary behaviors (OR = 1.43; CI = 1.21–1.69) were positively associated with being involved in fighting behavior.

## DISCUSSION

The present study contributes to knowledge on the subject of youth violence among Egyptian youth by examining social and demographic correlates for physical fighting. Study findings may be particularly useful and timely within the context of recent political changes in Egypt and the surrounding region; the charged atmospheres that accompany

**Table 4 Multivariate analysis of physical fighting among school-attending adolescents in Egypt (2006).**

| Variable | OR | 95% CI | *p*-value |
|---|---|---|---|
| Age | 0.96 | 0.89–1.02 | 0.232 |
| Sex (female) | 0.41 | 0.36–0.46 | <0.001 |
| Food insecurity | 1.64 | 1.42–1.91 | <0.001 |
| Bullied | 2.77 | 2.41–3.17 | <0.001 |
| Loneliness | 1.71 | 1.39–2.09 | <0.001 |
| Anxiety | 1.72 | 1.39–2.11 | <0.001 |
| Signs of depression | 1.26 | 1.10–1.45 | 0.001 |
| Truancy | 1.51 | 1.27–1.78 | <0.001 |
| Social network | | | |
|    No close friends | 0.87 | 0.66–1.15 | 0.341 |
|    One close friend | 0.78 | 0.63–0.95 | 0.341 |
|    Two close friends | 1.07 | 0.91–1.25 | 0.390 |
|    Three or more close friends | — | — | — |
| Helpful peers | 0.63 | 0.54–0.74 | <0.001 |
| Sedentary | 1.65 | 1.42–1.91 | <0.001 |
| Physically active | 0.91 | 0.78–1.08 | 0.314 |
| Understanding parents | 0.67 | 0.56–0.81 | <0.001 |

**Notes.**

OR, Odds Ratio; 95% CI, 95% Confidence Interval.
All estimates are adjusted for age and sex; age; or sex.

political revolutions and lack of safety measures in schools may exacerbate existing youth violence, likely making it an emergent public health issue in the region.

Overall our analyses yielded results that coincide with previously known risk and protective factors for youth involvement in interpersonal violence. Additionally, the reported prevalence rate of physical fighting among our sample of Egyptian adolescents (31%) was similar to reported rates in high-income (*Eaton et al., 2012*; *Fraga et al., 2011*) and other low- and middle-income countries globally (*Alikasifoglu et al., 2004*; *Rudatsikira, Muula & Siziya, 2008*). Well-studied risk factors such as signs of depression, truancy, and bullying victimization were associated with involvement in physical fighting in our sample of Egyptian adolescents. The commonly and globally observed trend of males being more involved in physical fighting was also found in our study (*Sousa et al., 2010*; *Walsh et al., 2013*). Protective factors such as understanding parents and helpful peers were also found to be significant for Egyptian adolescents. Two of our findings were notable for their novelty within the context of the more researched risk and protective factors for youth violence.

The finding of sedentary behavior's association with involvement in physical fighting adds to a small body of literature on physical activity and sedentary behavior and their effect on violent behavior in youth (*Nelson & Gordon-Larsen, 2006*; *Iannotti et al., 2009*; *Morris & Johnson, 2010*). As evidenced by the paucity of extant literature on the topic, the association between level of physical activity and involvement in physical fighting has not been extensively researched, but repeated significant findings among these handfuls of

**Table 5 Multivariate analysis of physical fighting among school-attending adolescents in Egypt (2006).**

| Variable | OR | 95% CI | *p*-value |
|---|---|---|---|
| Age | 0.90 | 0.84–0.97 | 0.007 |
| Sex (female) | 0.38 | 0.32–0.43 | <0.001 |
| Food insecurity | 1.00 | 0.73–1.37 | 0.982 |
| Bullied | 2.44 | 2.12–2.83 | <0.001 |
| Loneliness | 1.17 | 0.92–1.49 | 0.200 |
| Anxiety | 1.25 | 0.99–1.59 | 0.063 |
| Signs of depression | 1.29 | 1.11–1.50 | 0.001 |
| Truancy | 1.26 | 1.05–1.52 | 0.014 |
| Social network | | | |
|     No close friends | 0.76 | 0.56–1.03 | 0.077 |
|     One close friend | 0.75 | 0.60–0.92 | 0.007 |
|     Two close friends | 1.04 | 0.89–1.22 | 0.645 |
|     Three or more close friends | — | — | — |
| Helpful peers | 0.75 | 0.62–0.90 | 0.002 |
| Sedentary | 1.43 | 1.21–1.69 | <0.001 |
| Physically active | 0.94 | 0.78–1.13 | 0.526 |
| Understanding parents | 0.86 | 0.69–1.06 | 0.153 |

**Notes.**

OR, Odds Ratio; 95% CI, 95% Confidence Interval.
All estimates are adjusted for all variables listed in the table.

studies suggest this potential risk factor may deserve more attention from public health researchers.

Of particular importance is determining what about sedentary behavior contributes to violent behavior. Current hypotheses center on the type of sedentary activities youth are engaging in, with emphasis on video and computer gaming. Exposure to video or computer games with violent content has been linked to aggression (*Anderson, 2004*). Gaming, however, is not as ubiquitous in Egypt as it is in high-income countries; the high prices of gaming consoles and games make them unattainable to the majority of the population: access is often limited to public gaming centers (*Garratt, 2012*). Therefore, the gaming hypothesis is likely irrelevant to Egyptian youth. A more likely explanation for this correlation in Egyptian adolescents lies in the current political situation in Egypt. The contentious and highly-charged political atmosphere could be forcing sedentary behavior upon this age group: studies show that adolescents living in areas characterized by turmoil are more likely to engage in sedentary behavior (*Kim et al., 2010*; *Carson & Janssen, 2012*). While this does not explain the nature of the interaction between sedentary behavior and physical fighting, it does provide a potential starting point for addressing this behavior in Egyptian adolescents in a culturally-sensitive manner, most notably creating opportunities for adolescents to engage in appropriate physical activities in a safe environment.

The protective effect of having fewer friends–and, more specifically, having only one friend–seems counterintuitive, especially considering that loneliness (also a symptom

of depression) was found to be associated with increased likelihood of participating in physical fighting in our sample. However, perhaps having fewer friends, and thus, a smaller social circle, reduces the likelihood of youth becoming exposed to other youth who may be involved in risky and delinquent behaviors; associating with delinquent peers is a known risk factor for youth violence (*Haynie, 2001*; *Haynie, 2002*). Another explanation for this finding could be the quality of the youths' friendships: fewer, but more supportive and exemplary friends could have a more positive effect on youth than a wider yet less meaningful social network.

While not a specific aim of the present study, the high rates of reported signs of depression and bullying victimization give cause for considerable concern, particularly since both were found to be strongly associated in our sample–at all levels of analysis–for involvement in physical fighting. An explanation for the high rates of reported signs of depression could be that most Egyptian adolescents are without hope for a bright future with appropriate education, descent work and marriage opportunities. Our findings would suggest that culturally-appropriate tertiary interventions designed to alleviate depressive symptoms and help adolescents cope with bullying victimization may help reduce the likelihood of involvement in physical fighting. In conjunction with these efforts, primary and secondary prevention programs designed to prevent and reduce bullying would be advantageous as well and most likely create supportive school environments; this may increase the likelihood of having helpful peers, which was an observed protective factor for involvement in physical fighting.

## Strengths and limitations

Demographic, socioeconomic and political factors singular to LMICs create a contextual backdrop unlike that in high-income countries. In particular, the "youth bulge" phenomenon observed in LMICs has been cited as a possible explanation for escalating violence in this demographic region (*Hilker & Fraser, 2009*). This disproportionate age distribution puts additional strains on underfunded infrastructures that may already struggle to provide employment and adequate education to the younger generation. Combined with political unrest, a precarious national ethos materializes that often translates to increased interpersonal conflicts and violence (*Peden et al., 2008*). Additionally, the effects of urbanization have led to increased migration from urban areas to surrounding rural areas, creating pockets of impoverished, suburban populations – situations which have been shown to increase aggressive and violent behavior among adolescents (*Schneiders et al., 2003*; *Carlson, 2006*).

The large sample size from a population within a country that is experiencing the aforementioned phenomena serves as a major strength of our study. Our findings therefore may be more generalizable to other LMICs than the current body of literature on risk and protective factors for adolescent interpersonal violence–much of which has been based on research done in HICs–and other studies in the emergent literature on adolescent interpersonal violence in LMICs that have used smaller sample sizes (*Ez-Elarab et al., 2007*).

Our findings, however, should be considered within the context of the study's limitations. Given the nature of self-report surveys, responses could be biased towards socially accepted norms. Some questions required participants to recall information within a rather generous time frame (30 days or 12 months), and the effects of the passage of time on memory might have affected participants' recall ability and accuracy. Furthermore, while designed to be culturally-sensitive and given to participants in their native language (Arabic), participant interpretation of the questions might have differed from what the survey designers intended to capture with the questions. Additionally, causality cannot be inferred due to the cross-sectional data.

### Future research

As mentioned earlier, more research is needed to understand why participation in sedentary behaviors seems to increase the likelihood of involvement in physical fighting, particularly in youth from low- and middle-income countries. Some professionals theorize that sedentary behavior often coincides with unstructured socializing, of which there has been an observed association with delinquent behaviors such as violence (*Haynie & Osgood, 2005*). This might be a more plausible explanation for the association between sedentary behavior and physical fighting in our sample of Egyptian youth, rather than gaming-centered hypotheses.

Or does sedentary behavior interact with other present risk factors such as depression? Depressive symptoms and a sedentary lifestyle are closely related; a standing chicken-and-the-egg debate over the nature of the association exists, but a growing body of literature indicates that targeted physical activity interventions may ameliorate depressive symptoms in adolescents (*Brown et al., 2013*). Involving Egyptian adolescents with a high risk for physical fighting in physical activities as part of a well-designed intervention program–and determining this program's effects–may be a likely next step for public health professionals working to address adolescent interpersonal violence in Egypt.

Studies designed to repeat the significant finding of fewer friends as a protective factor should also be conducted. Quantitative and qualitative studies could probe deeper into the nature of the friendships to learn which specific qualities of the relationship impart protective effects. Once more is known about this potential protective factor, public health professionals will have a better understanding of how to include it in intervention efforts, if the research does indeed suggest it should be taken into consideration.

### CONCLUSIONS

Little is known about the epidemiology of youth violence in Egypt. Our research adds insight into this public health issue, reporting trends in physical fighting among Egyptian adolescents. Prevalence rates appear to be similar to those reported in high-income countries and other low- and middle-income countries. The well-known risk factors of depression and bullying victimization are prominent among Egyptian adolescents; while the lesser understood risk factor of sedentary behavior is present as well. Protective factors center on social aspects such as understanding parents and helpful peers. Interventions aimed at attenuating depressive symptomatology and creating supportive school

environments, as well as providing access to physical activities in a safe environment, may help reduce the prevalence of violent behavior in Egyptian adolescents.

## ACKNOWLEDGEMENTS

The authors thank the Ministry of Education, Arab Republic of Egypt; the Centers for Disease Control and Prevention (Atlanta, GA, USA); and the World Health Organization (Geneva, Switzerland). The authors also thank all of the survey officers and students who took part in the survey.

### Funding

The authors declare that they have no funding for this work.

### Competing Interests

The authors declare that they have no conflict of interest, whether financial or otherwise, which arise from the analysis and reporting of the data contained in this manuscript.

### Author Contributions

- Karen L. Celedonia conceived and designed the experiments, wrote the paper.
- Michael L. Wilson performed the experiments, analyzed the data, wrote the paper.
- Hanan A. El Gammal reviewed and made critical revisions to the manuscript.
- Abeer M. Hagras contributed reagents/materials/analysis tools.

### Ethics

The following information was supplied relating to ethical approvals (i.e., approving body and any reference numbers):

Egypt Ministry of Education research committee

The study was based on freely available secondary data. Detailed information on ethical procedures and policies concerning the data collection can be found at http://www.cdc.gov/gshs/.

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
