# Peer review of "Physical fighting among Egyptian adolescents: social and demographic correlates among a nationally representative sample"

_PeerJ, doi:10.7717/peerj.125_

## Round 0.1 · original submission · Major Revisions

· Academic Editor

Major Revisions

This paper deals with an interesting topic, and number of respondents is enough to be representative of the national adolescent behavior. Design of study is appropriate. It needs some amendments before to be accepted to be published in PeerJ.

Major.
Data used in this study is quite old (2006). Since then several social, economic and political changes have been occurred in Egypt. I suggest actualizing data to better reflect present social situation in Egypt I order to better understand situation of Egyptian adolescents. This opinion is supported by the fact that the authors used actualize references (i.e.: 2012, 2013), and therefore is quite difficult to accept present results obtained at so long time ago from the present time.

Minor.

Introduction is too long. It must be shortened at least to 1 and half page. Last 4 lines after the declaration of aims (from “Study findings…” to the end) could be deleted and used in the discussion, avoiding repetitions.

Methods.
Authors must state why the student response rate is 87%.
Authors also stated that this is a self-reported study. Did the authors some validation to be sure that the self-reported method is yields appropriate results? If authors use a validated methods, please, give appropriate reference.

Discussion
Authors must declare in limitations of the study that this is a cross-sectional study, and then no relationships cause-effect can be done.

Reviewer 1 ·

Basic reporting

This paper is interesting since epidemiological data on individual and social correlates of physical violence in Egyptian adolescents is limited. The results highlight high prevalence rates of adolescent interpersonal violence in Egypt. High rates of depressive symptomatology and bully victimization along with their positive association with physical fighting suggest that interventions aimed at treating and preventing these problems may help mitigate the likelihood of adolescents engaging in violent behavior; involvement in appropriate physical activity in a safe environment may be beneficial as well.
The manuscript is well written and structured. The hypothesis and aims are also clearly stated. However, some minor modifications should be made.

Experimental design

No Comments

Validity of the findings

No Comments

Additional comments

Introduction
1. “40%” instead of “40 percent”; and “70%” instead of “70 percent”.
2. The manuscript would benefit is the authors could include page 3 of the introduction section, lines 1-4 and lines 7-11 in a “limitations and strengths” section.
Methods
1. The manuscript would benefit if the authors include a table detailing the information on the data variables created.
2. The manuscript would benefit if the authors detail “anxiety and signs of depression” as individual-level variables separately.
3. Please, remove “for Linux (www.linuxmint.com)”.
Results
1. Page 1 of the “Results” section: “The mean age of the sample was 13.2 years old” instead of “The mean age of the sample was 13.2 years”.
2. Why the authors included “food insecurity” without discussing this variable?
3. Please, change “Table 3” for “Table 2”.
4. The manuscript would benefit including a table assessing males and females separately since females were less likely than males to be involved in physical fights.

---

## Round 0.2 · accepted · Accept

· Academic Editor

Accept

Thanks to consider editor and referee suggestions. We'll remain waiting for new papers from yours.